# Alcohol dependence promotes systemic IFN-γ and IL-17 responses in mice

**Kayla Frank**[1], **Shawn Abeynaike**[1], **Rana Nikzad**[1], **Reesha R. Patel**[2], **Amanda J. Roberts**[3], **Marisa Roberto**[2], **Silke Paust**[1]*

**1** Department of Immunology and Microbiology, The Scripps Research Institute, La Jolla, CA, United States of America, **2** Department of Molecular Medicine, The Scripps Research Institute, La Jolla, CA, United States of America, **3** Animal Models Core, The Scripps Research Institute, La Jolla, CA, United States of America

* paust@scripps.edu

**Data Availability Statement:** All relevant data are within the manuscript and its Supporting Information files.

**Funding:** Support for this study was provided by the National Institute on Alcohol Abuse and Alcoholism grants AA021491, AA013498, P60

## Abstract

Alcohol use disorder (AUD) is a chronic relapsing disorder characterized by an impaired ability to stop or control alcohol use despite adverse social, occupational, or health consequences. AUD is associated with a variety of physiological changes and is a substantial risk factor for numerous diseases. We aimed to characterize systemic alterations in immune responses using a well-established mouse model of chronic intermittent alcohol exposure to induce alcohol dependence. We exposed mice to chronic intermittent ethanol vapor for 4 weeks and analyzed the expression of cytokines IFN-γ, IL-4, IL-10, IL-12 and IL-17 by different immune cells in the blood, spleen and liver of alcohol dependent and non-dependent control mice through multiparametric flow cytometry. We found increases in IFN-γ and IL-17 expression in a cell type- and organ-specific manner. Often, B cells and neutrophils were primary contributors to increased IFN-γ and IL-17 levels while other cell types played a secondary role. We conclude that chronic alcohol exposure promotes systemic pro-inflammatory IFN-γ and IL-17 responses in mice. These responses are likely important in the development of alcohol-related diseases, but further characterization is necessary to understand the initiation and effects of systemic inflammatory responses to chronic alcohol exposure.

## Introduction

Chronic intermittent alcohol exposure is associated with increased risk of cancer, organ damage, and infection [1–3]. At the molecular level, chronic alcohol alters inflammatory processes which regulate immune function. This is often characterized by changes in cytokine expression and immune defense mechanisms of type 1—cell-mediated immunity associated with IFN-γ, IL-12 and TNF-α expression—or type 2—humoral immunity associated with IL-4, IL-10 and IgE expression—immune responses [4]. One mechanism through which chronic alcohol exposure leads to adverse clinical outcomes is well characterized: increased expression of TNF-α by liver macrophages promotes inflammation leading to alcoholic liver disease (ALD) development in human alcoholics [5,6].

AA006420, AA017447, AA015566, AA027700, F32
AA026765, T32 AA007456, as well as the Pearson
Center for Alcoholism and Addiction Research (all
to MR) and The Scripps Research Institute's
unrestricted funds (SP). The manuscript is The
Scripps Research Institute's MS# 30018.

**Competing interests:** The authors have declared
that no competing interests exist.

Work in humans and rodents demonstrates that altered TNF-α expression is not limited to the liver; TNF-α levels also increase in the blood and spleen following chronic alcohol exposure [7,8]. Furthermore TNF-α is not the only cytokine which exhibits increased expression following chronic alcohol exposure. IFN-γ and IL-17, among others, are increased in systemic pro-inflammatory responses [9–12]. Notably, IL-17 is not a type 1 or type 2 cytokine; it mediates highly inflammatory type 17 responses [13,14]. Macrophages may play a central role in TNF-α production following chronic alcohol exposure, but a variety of other immune cell types—B cells, T cells, NK cells, NKT cells, neutrophils and dendritic cells—have also been implicated as mediators of alcohol-associated inflammation [10,11,15–17].

Despite the numerous studies addressing organ-specific inflammatory response mechanisms of individual cell types to chronic alcohol exposure, the field lacks knowledge of the interplay between these responses systemically. Furthermore, most studies analyze differences in the immune response following chronic alcohol exposure in the context of infection or disease progression. Less is known about the effects of chronic alcohol on immune function prior to infection or disease development.

Thus, in this study, we aimed to characterize changes in the immune system of chronic ethanol exposed mice (alcohol dependent mice—[18–23]) at steady state through multiparametric flow cytometry. Looking beyond TNF-α's established functions, we analyzed changes in other pro- and anti-inflammatory cytokines—IFN-γ, IL-4, IL-12, IL-17 and IL-10—and the cells which produce them—B cells, T cells, NK cells, NKT cells, macrophages, neutrophils and dendritic cells—in the blood, spleen and liver of alcohol dependent mice. We found significant systemic upregulation of IFN-γ and IL-17 and identified B cells and neutrophils as major contributors to these responses. The results add new insight to the systemic effects of alcohol dependence on immunity and the organ-specific properties of the response.

## Materials and methods

### Animals

Male IL10[tm1Flv] (stock no: 008379) mice were obtained from The Jackson Laboratory (ME) and bred in-house and used to visualize IL-10 production by various immune cells using gfp. IL10[tm1Flv] mice are a reporter strain used to detect and monitor cells committed to interleukin 10 (IL10) production and recapitulate a normal IL-10 expression pattern [24]. Mice were group-housed in a temperature and humidity-controlled vivarium on a 12 hour reversed light/ dark cycle with food and water available *ad libitum*. All protocols involving the use of experimental animals in this study were approved by The Scripps Research Institute's Institutional Animal Care and Use Committee and were consistent with the National Institutes of Health Guide for the Care and Use of Laboratory Animals.

### Chronic-intermittent ethanol vapor exposure

To induce ethanol dependence, mice were exposed to chronic intermittent ethanol inhalation as previously described [18–20,22,23,25]. Briefly, mice in the dependent group (n = 8) were i. p. injected with 1.75 g/kg alcohol + 68.1 mg/kg pyrazole (alcohol dehydrogenase inhibitor) and placed in vapor chambers (La Jolla Alcohol Research, La Jolla, CA) for 4 days (16 hours vapor on, 8 hours off) followed by 72 hours of forced abstinence [20,26]. This regimen was repeated for a total of 4 full rounds. Non-dependent mice (n = 5) were injected with 68.1 mg/ kg pyrazole in saline and received only air in similar chambers for the same intermittent period as the dependent group. On the third day of vapor exposure tail blood was collected to determine blood ethanol levels (BELs). Alcohol drip rates in the vapor chambers were altered such that BELs progressively increased over the vapor rounds to a final target of 200–250 mg/

dL. As in our previous studies [18–20,22,23], dependent mice were euthanatized right after the last alcohol vapor exposure (while there were still intoxicated).

## Cell isolation and flow cytometry analysis

Single cell suspensions were generated from blood, spleen, and liver of alcohol dependent and non-dependent control mice. Cells were extracted from spleen and livers by mechanical disruption and washed with PBS through 40-μm mesh filters. Immune cells were enriched through incubation with ACK buffer (spleen) or through density gradient centrifugation using Ficoll-Paque (GE Healthcare) following the manufacturer's protocol (blood and liver). Immune cells were washed with PBS containing 2% FBS and incubated with murine Fc Block at 4° C for 10 min prior to staining. Cells were stained with extracellular antibodies, washed, permeabilized, incubated with Fc Block again, and finally stained with intracellular antibodies. Permeabilization and intracellular staining steps were performed with the FoxP3 permeabilization buffer kit (Tonbo Biosciences, San Diego, CA) following the manufacturer's protocol. Flow cytometry data was acquired on a four-laser Aurora (Cytek, Fremont, CA) and analyzed with FlowJo v 10.6.2 (Becton, Disckinson, and Company, Franklin Lakes, NJ). Graphing and statistical analyses were performed on Graphpad Prism 8.4.0 (San Diego, CA). S1 Table provides a list of the antibodies used for the experiment.

## Results

### Systemic immune responses are altered in alcohol dependent mice

We observed organ-specific changes in immune cell and cytokine expression in alcohol dependent mice compared to non-dependent controls. S1 Fig summarizes the gating strategy used to identify the various CD45[+] cell types highlighted throughout this analysis. We present cell type expression as percentages of total CD45[+] cells; an increase does not always translate to higher cell numbers but could be due to decreases in other cell types and less total CD45[+] cells. B cells, NK cells, T cells and NKT cells represent a higher proportion of CD45[+] cells than macrophages, neutrophils and dendritic cells in the blood, spleen and liver of dependent and non-dependent mice (Fig 1A–1C).

The data demonstrate significant organ-specific differences in the distribution of immune cells in alcohol dependent mice compared to non-dependent controls: the blood exhibits changes in both lymphocyte and myeloid cell populations, whereas the spleen is mostly impacted by altered myeloid cell expression and changes in the liver are limited to lymphocytes. Of note, the relative abundance of neutrophils increases significantly in the blood and the spleen (Fig 1A and 1B). The liver exhibits significant changes in lymphocyte expression: a large decrease in liver B cells is complemented by an increase in T cells and NKT cells (Fig 1C). The changes in blood lymphocyte populations did not reach significance, but the data resemble the dynamic changes of the liver more than the consistency of the spleen. Although, we did not investigate the cause of altered immune cell expression in alcohol dependent mice, proliferation, migration, or cell death could contribute to these changes. Furthermore, we do not present cell numbers because the total cells analyzed from each mouse varied. Fig 1A–1C primarily serve as a reference to contextualize cell specific-changes in the upcoming figures.

CD45[+] cells from the blood, spleen and liver also show differences in cytokine expression between dependent and non-dependent controls. We measured expression of type 1 cytokines IFN-γ and IL-12, type 2 cytokines IL-4 and IL-10, and type 17 cytokine IL-17. IFN-γ and IL-17 levels increase in all three organs, although the changes are not statistically significant in the liver. IL-12 expression did not change significantly in any organs despite its traditional role in stimulating IFN-γ expression [27]. We did observe a significant increase in IL-10 expression in the liver, but we question the physiological relevance because of its low frequency (Fig 1D–1F).

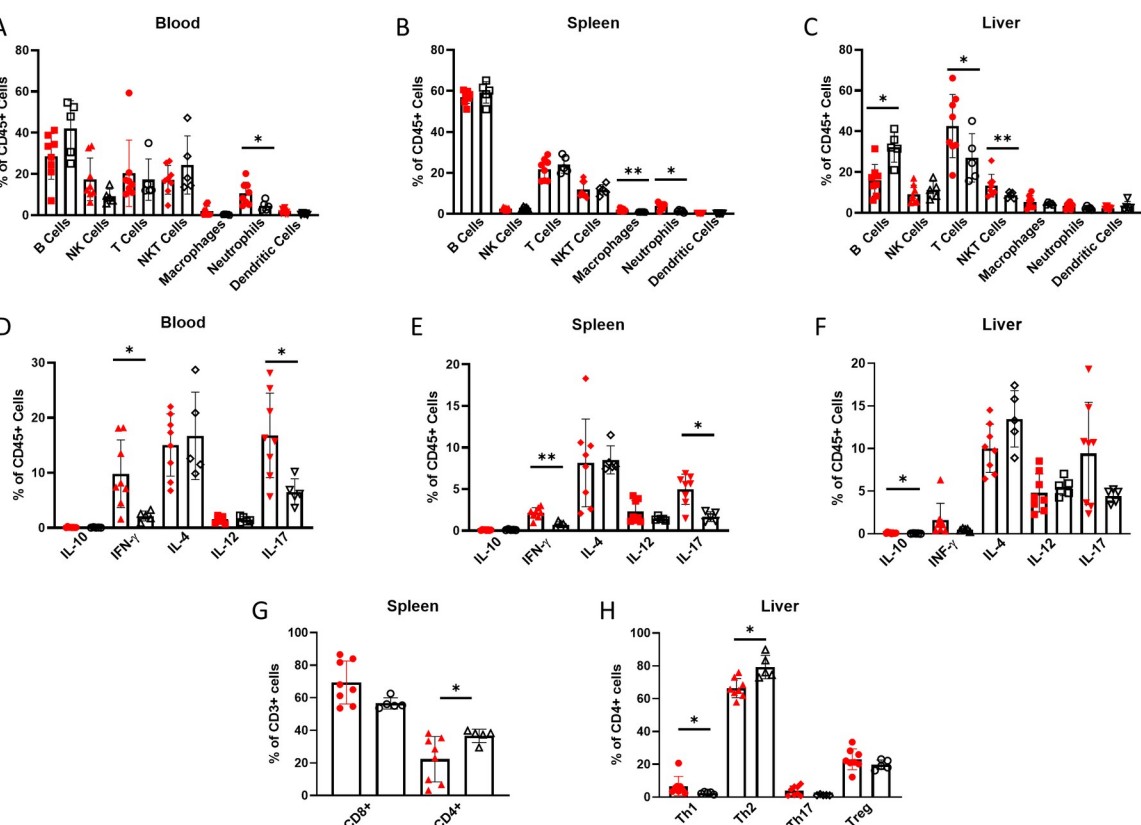

**Fig 1. Alcohol dependent and non-dependent mice present with significant differences in immune cell compositions and cytokine expression.** (A-C) Immune cell expression as percentages of all CD45⁺ cells in (A) blood, (B) spleen and (C) liver isolated from alcohol dependent (red, solid fill) and non-dependent (black outline, white fill) mice. (D-F) Expression of cytokines in (D) blood, (E) spleen and (F) liver CD45⁺ cells isolated from dependent and non-dependent mice. (G) CD4⁺ and CD8⁺ T cell expression as a percentage of CD3⁺ cells isolated from spleen of dependent and non-dependent mice. (H) Th1 (CD4⁺ IFN-γ⁺), Th2 (CD4⁺ IL-4⁺), Th17 (CD4⁺ IL-17⁺), and Treg (CD4⁺ FoxP3⁺) expression as a percentage of CD4⁺ T helper cells isolated from liver of dependent and non-dependent mice.*, p<0.05; **, p<0.01 analyzed by Mann-Whitney U test; n = 5–8.

Nevertheless, we found strong evidence that chronic alcohol exposure promotes systemic IFN-γ and IL-17 responses.

Next, we investigated further into expression of CD4⁺ T cell subsets which selectively express these cytokines: Th1, Th2, Th17 and T regulatory (Treg) cells. They are often considered the main regulators of systemic type 1, 2, and 17 immunity and IL-10-mediated regulatory responses, respectively. We found reduced CD4⁺ T cell expression in the spleen of alcohol dependent mice, but we did not observe significant differences in the cytokine expression profile of splenic CD4⁺ T cells (Figs 1G and S2A). Conversely, we did not see a difference in CD4⁺ and CD8⁺ T cell expression in the liver, but observed a clear increase in the Th1/Th2 ratio (Figs 1H and S2B). Despite their increase, Th1 cells in the liver are expressed at low frequency and are likely not the only contributor to increased IFN-γ levels overall. We found minimal differences in CD4⁺ T cell cytokine expression between the two groups in the spleen (S2C and S2D Fig). Although we observed clear increases in overall IFN-γ and IL-17 levels, we do not see large changes in CD4⁺ T cell expression patterns. It is unlikely that changes in T cell cytokine expression alone are driving these type 1 and type 17 immune responses. Therefore, we hypothesized that other immune cell types are important in promoting type 1 and type 17 immunity in alcohol dependent mice.

## Pro-inflammatory type 1 and type 17 responses are altered in alcohol dependent mice

We analyzed the source of increased pro-inflammatory IFN-γ and IL-17 cytokine responses in alcohol dependent mice compared to non-dependent controls. First, we analyzed the expression of IFN-γ as a percentage of each individual cell type in all three organs (Fig 2A–2G). We also investigated the cell types responsible for total IFN-γ levels in the blood, spleen and liver (Fig 2H–2J). Together, these analyses give a better understanding of the source(s) of increased IFN-γ expression in each organ. Interestingly, the proportions of cell types which make up the total CD45$^+$ IFN-γ$^+$ population do not mirror the distribution of all CD45$^+$ cells. For example, macrophages and neutrophils are among the largest contributors to IFN-γ expression whereas they make up a minimal proportion of CD45$^+$ cells for all three organs (Figs 1A–1C and 2H–2J).

In the blood, the percent of B cells and neutrophils which produce IFN-γ is increased significantly; B cell expression increases more dramatically than neutrophil expression, but a much higher percentage of neutrophils produce IFN-γ overall (Fig 2A and 2F). The distribution of

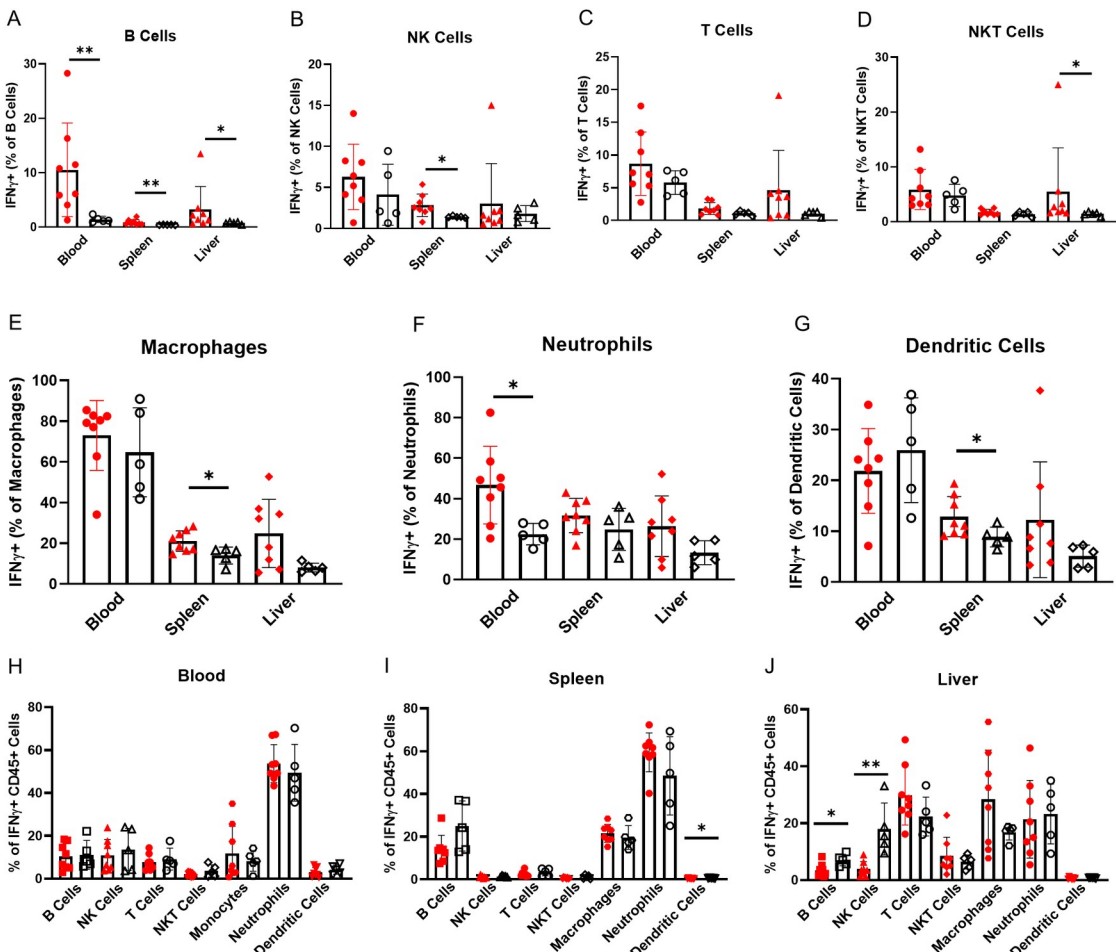

**Fig 2. IFN-γ expression is increased in alcohol dependent compared to control mice.** (A-G) Expression of IFN-γ by (A) B cells, (B) NK cells, (C) T cells, (D) NKT cells, (E) macrophages, (F) neutrophils and (G) dendritic cells in alcohol dependent (red, solid fill) and non-dependent (black outline, white fill) mice. (H-J) Immune cells as percentages of all IFN-γ producing CD45$^+$ cells in (H) blood, (I) spleen and (J) liver isolated from dependent and non-dependent mice. *, p<0.05; **, p<0.01 analyzed by Mann-Whitney U test; n = 5–8.

CD45$^+$ IFN-γ$^+$ cell types are mostly consistent between dependent and non-dependent mice. Neutrophils are responsible for 50–60% of IFN-γ production in both groups (Fig 2H). Neutrophils are likely the main source of increased IFN-γ in the blood of the alcohol dependent mice, while B cells are a secondary source.

We also found that splenic B cells are characterized by increased IFN-γ expression (Fig 2A); the expression level is low, but B cells comprise about 60% of total CD45$^+$ cells and 20–30% of CD45$^+$ IFN-γ$^+$ cells in the spleen (Figs 1B and 2I). Increased IFN-γ expression in splenic macrophages complements the increase in B cell IFN-γ expression (Fig 2E). Despite representing a smaller percentage of CD45$^+$ cells, they contribute similarly to total CD45$^+$ IFN-γ$^+$ cells because about 20% of them produce IFN-γ (Figs 1B and 2I). NK cells and DCs also show increased expression of IFN-γ in the spleen, although their minimal contributions to total IFN-γ levels may render this change less important to overall IFN-γ responses (Fig 2B, 2G and 2I).

The role of neutrophils is unclear in the spleen. They do not show increases in IFN-γ expression on a per cell basis (Fig 2F), however they increase as a proportion of CD45$^+$ IFN-γ$^+$ cells in the spleen in dependent mice (Fig 2I). This could be explained by their significant increase as a percentage of CD45$^+$ cells (Fig 1B): increased abundance of neutrophils contributes to increased IFN-γ expression without significant changes in the percent of neutrophils producing IFN-γ. Increases in spleen B cell and macrophage IFN-γ levels—and potentially increases in neutrophil abundance—are the main source of observed increases in IFN-γ in the spleen of alcohol dependent mice.

Dynamic changes in IFN-γ expression patterns occur in the liver. Liver B cells and NKT cells have significant increases in IFN-γ expression (Fig 2A and 2D), however non-significant trends towards increased IFN-γ in T cells, macrophages and neutrophils (Fig 2C, 2E and 2F) should be considered due to their higher relative contributions to the total IFN-γ pool in the liver. NK cells make up about 20% of IFN-γ producing cells in non-dependent mice but become a strikingly lower contributor to total IFN-γ$^+$ cells in dependent mice. This is accompanied by trends towards decreases in B cell and increases in T cell and macrophage contributions to total IFN-γ levels (Fig 2J). Although the trend towards increased total expression of IFN-γ in the liver was not significant (Fig 1F), the data here show a variety of changes in the production of IFN-γ across nearly every cell type. Altogether, **Fig 2** demonstrates that activation of the type 1 immune response is a prominent aspect of chronic alcohol exposure's immunomodulatory effects: the blood, spleen and liver each exhibit activation of type 1 responses in an organ-specific manner.

We then completed a parallel analysis to understand increased IL-17 expression in alcohol dependent mice compared to non-dependent controls (Fig 1D–1F). We investigated the expression of IL-17 by various cell types in the blood, spleen and liver (Fig 3A–3G). Further, we analyzed the distribution of all CD45$^+$ IL-17$^+$ cells to understand the significance of these cell type-specific changes (Fig 3H–3J). Like CD45$^+$ IFN-γ$^+$ cells, CD45$^+$ IL-17$^+$ cells do not mirror the distribution of all CD45$^+$ cells: macrophages and neutrophils represent a much larger proportion of IL-17 producing CD45$^+$ cell than all CD45$^+$ cells (Figs 1A–1C and 3H–3J).

In the blood of alcohol dependent mice, the percentage of B cells, NK cells and neutrophils expressing IL-17 increases significantly (Fig 3A, 3B and 3F). The B cell population increases significantly as a percentage of all CD45$^+$ IL-17$^+$ cells, accompanied by trends towards increasing NK cell and decreasing NKT cell percentages. Neutrophils are the dominant CD45$^+$ IL-17$^+$ cell type in both alcohol dependent mice and non-dependent controls (Fig 3H). Of the lymphocytes, NK cells may be more important contributors to increased IL-17 levels than B cells: NK cell expression levels increase in the blood of dependent mice whereas B cells decrease (Fig 1A) and NK cells contribute more significantly to the total blood CD45$^+$ IL-17$^+$ cell population

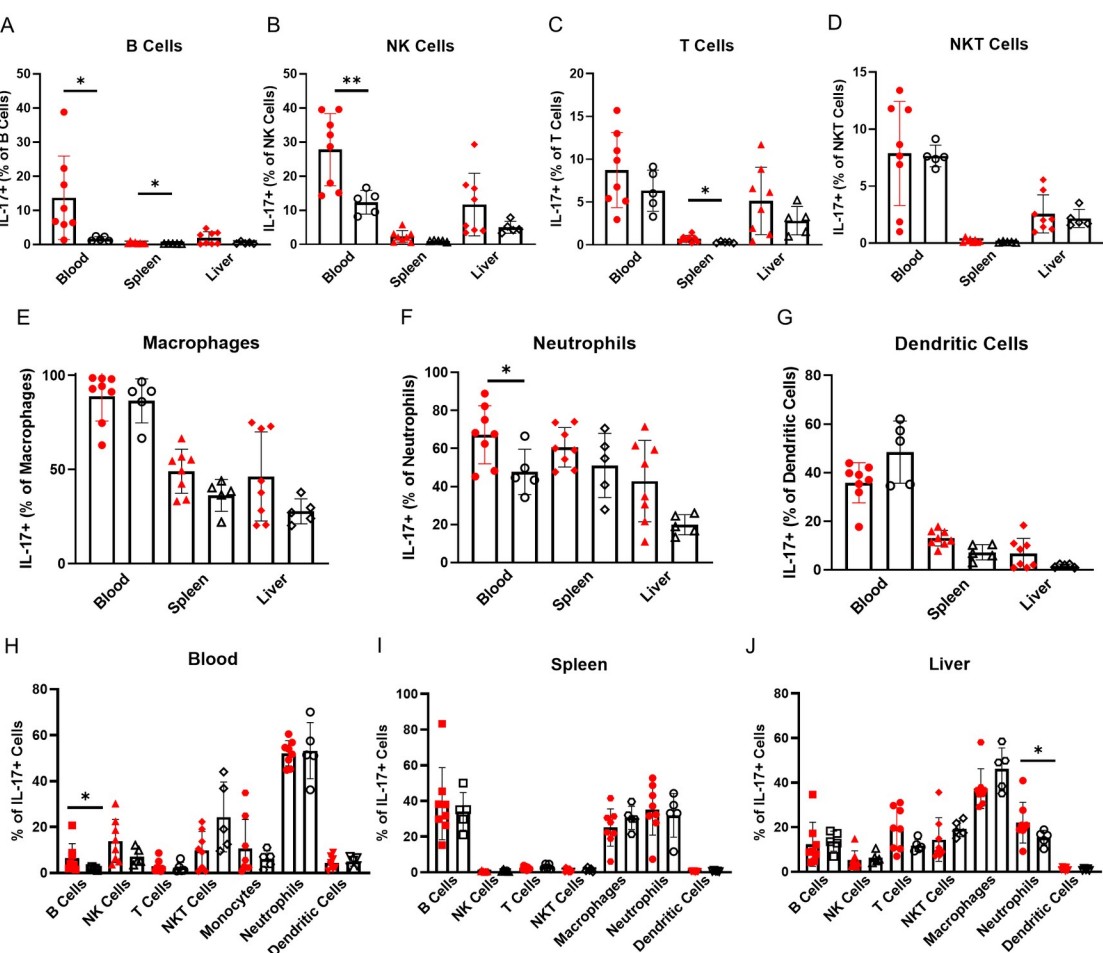

**Fig 3. IL-17 expression is increased in alcohol dependent compared to control mice.** (A-G) Expression of IL-17 by (A) B cells, (B) NK cells, (C) T cells, (D) NKT cells, (E) macrophages, (F) neutrophils and (G) dendritic cells in alcohol dependent (red, solid fill) and non-dependent (black outline, white fill) mice. (H-J) Immune cells as percentages of all IL-17 producing CD45+ cells in (H) blood, (I) spleen and (J) liver isolated from dependent and non-dependent mice. *, p<0.05; **, p<0.01 analyzed by Mann-Whitney U test; n = 5–8.

(Fig 3H). NK cells, B cells and neutrophils together are the major sources of increased IL-17 expression in the blood of dependent mice.

B cell IL-17 expression increases significantly in the spleen of alcohol dependent mice, although the frequency of IL-17+ B cells is quite low (Fig 3A). Like for IFN-γ, this is counter-balanced by an abundance of B cells in the spleen (Fig 1B); a small increase in the percent of IL-17+ splenic B cells could still promote type 17 responses. B cells, along with macrophages and neutrophils, make up the vast majority of CD45+ IL-17+ cells; this distribution does not vary between dependent and non-dependent groups (Fig 3I). Non-significant trends towards increased IL-17 expression in macrophages and neutrophils are likely important in the overall increases in IL-17 observed in the dependent spleen (Fig 3E and 3F). T cells also show significantly increased IL-17 expression (Fig 3C), but the frequency is low—around 2% in dependent mice—and the contribution of T cells to total IL-17+ CD45+ cells is minimal (Fig 3I).

No specific cell populations in the liver show significant increases in IL-17 expression, so it is difficult to attribute the overall increase in IL-17+ cells to one cell type (Fig 3A–3G). However, each of them exhibits a trend towards increasing IL-17 expression which likely add up to

the trend of increased in IL-17 levels in the liver (Fig 1F). The proportion of neutrophils in the CD45$^+$ IL-17$^+$ population increases significantly. T cells also show increasing trends which are offset by decreased contributions by NKT cells, B cells and macrophages to total IL-17$^+$ cells (Fig 3J). Despite a potential decrease in macrophages, they still represent the largest proportion of CD45$^+$ IL-17$^+$ cells in the liver. Trends towards increased IL-17 expression across numerous cell types, considered with changes in the distribution of CD45$^+$ IL-17$^+$ cells, suggests a dynamic type 17 immune response in the liver: interactions between a variety of cell types contribute to excess IL-17 production and inflammation.

## Anti-inflammatory immune responses are not altered in alcohol dependent mice

We also observed a significant increase in IL-10 levels in the liver of alcohol dependent mice compared to non-dependent controls (Fig 1F). Although in both groups IL-10 is only expressed in about 0.1% of CD45$^+$ cells, we investigated the potential for increased anti-inflammatory responses in alcohol dependent mice. The abundance of IL-10 was too low to accurately analyze which cell types produce notable amounts, but we analyzed the expression of the IL-10 receptor (IL-10R) and found a significant increase in the spleen and a trend towards increased IL-10R expression in the blood of dependent mice compared to non-dependent controls (S3A Fig). It is unclear whether changes in IL-10R expression are important in this context as very few cells express IL-10 in the organs analyzed from both groups.

Nevertheless, we analyzed IL-10R expression as a percent of each cell type (S3B–S3H Fig) and the cell types which make up the total CD45$^+$ IL-10R$^+$ population (S3I–S3K Fig) in the blood, spleen and liver of alcohol dependent and non-dependent mice. The most notable change between the two groups is an increase in IL-10R expression as a percentage of total blood B cells (S3B Fig). This is reflected by a significantly higher expression of IL-10R as a percentage of total blood B cell (S3I Fig). We observe consistent increases in B cell activation in our data; increased IL-10R expression could be a measure to prevent overactivation, but with low levels of circulating IL-10 a higher IL-10R expression would have minimal effects.

In general, lymphocyte IL-10R expression in alcohol dependent mice increases whereas myeloid cell IL-10R expression decreases (S3B–S3H Fig). In line with increases in IL-10R expression blood, spleen, and liver lymphocytes, we also found increases in PD-1 expression in lymphocytes overall (S4A–S4D Fig). These increases are most pronounced in B cells, however trends towards increased expression are evident in other lymphocytes as well. Further investigation of anti-inflammatory responses and B cell regulation in response to chronic alcohol exposure would be beneficial to understand the physiological relevance of these changes.

## Co-expression of IFN-γ and IL-17 in alcohol dependent mice

We observed that the cell types responsible for both IFN-γ and IL-17 expression are distributed similarly in the blood, spleen, and liver (Figs 2H–2J and 3H–3J). Thus, we investigated this further by analyzing the co-expression of these inflammatory cytokines and the cell types which make them in the blood, spleen, and liver of alcohol dependent and non-dependent control mice (Fig 4A–4H). The percent of CD45$^+$ cells which co-express IFN-γ and IL-17 increases significantly in both the blood and liver of alcohol dependent mice compared to non-dependent controls. Co-expression of these cytokines in the spleen of dependent mice also trends towards an increase which is not significant (Fig 4A). The contribution of different cell types to total CD45$^+$ cells co-expressing IFN-γ and IL-17 differs from expression of each cytokine individually in one major aspect: B cells do not make up a large percentage of these

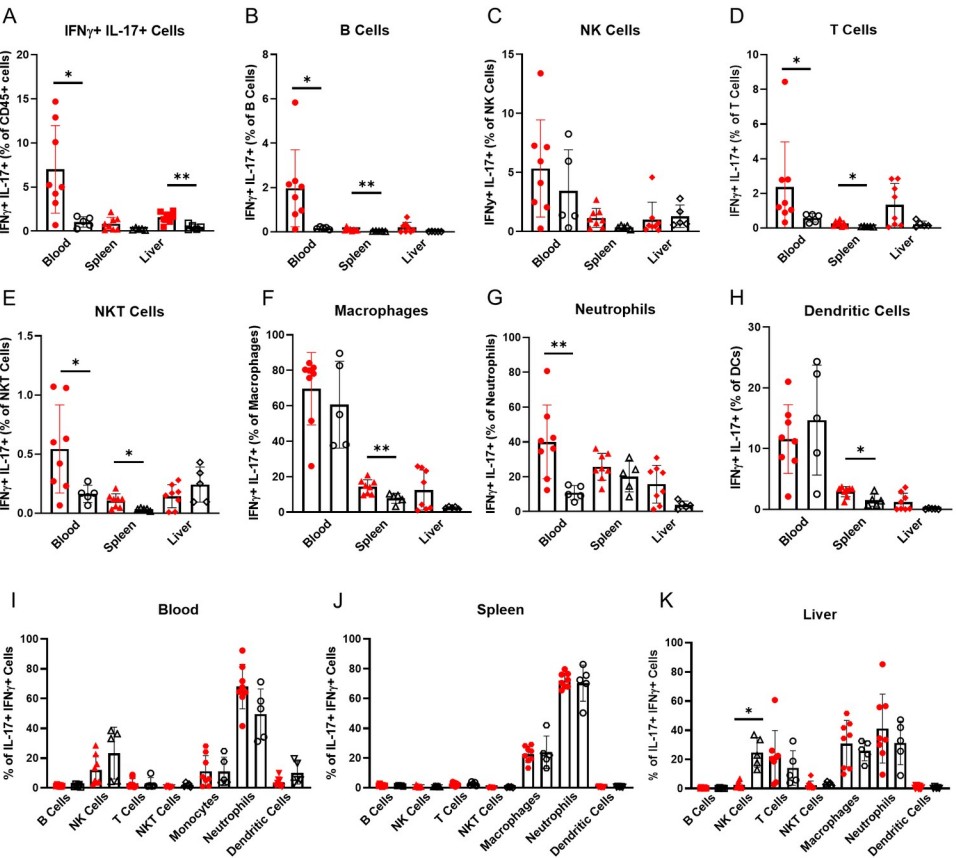

**Fig 4. Immune cells co-expressing IFN-γ and IL-17 are significantly increased in alcohol dependent mice.** (A) Co-expression of IFN-γ and IL-17 in blood, spleen, and liver CD45+ cells isolated from alcohol dependent (red, solid fill) and non-dependent (black outline, white fill) mice. (B-H) Expression of IL-17 by (B) B cells, (C) NK cells, (D) T cells, (E) NKT cells, (F) macrophages, (G) neutrophils and (H) dendritic cells in dependent and non-dependent mice. (I-K) Immune cells as percentages of all IFN-γ+ IL-17γ+ CD45+ cells in (I) blood, (J) spleen and (K) liver isolated from dependent and non-dependent mice. *, p<0.05; **, p<0.01 analyzed by Mann-Whitney U test; n = 5–8.

cells in any of the analyzed organs. Macrophages and neutrophils, however, remain prominent contributors to total cytokine expression levels (Fig 4I–4K).

In the blood—which sees the largest increase in IFN-γ and IL-17 co-expressing cells in dependent mice compared to non-dependent controls—B cells, T cells and NKT cells increase in percent co-expressing IFN-γ and IL-17 (Fig 4B, 4D and 4E). However, these cells co-express IFN-γ and IL-17 at a much lower frequency than neutrophils. Neutrophils also have significantly higher co-expression in dependent mice comparted to non-dependent controls and account for the highest percent of CD45+ IFN-γ+ IL-17+ cells (Fig 4G and 4I). NK cells—the next most abundant IFN-γ+ IL-17+ cell type—do not show significantly different co-expression of these cytokines between groups (Fig 4C and 4I). Neutrophils in the blood co-expressing IFN-γ and IL-17 could be a major instigator of inflammatory immune responses to chronic alcohol exposure.

Splenic macrophages, B cells, T cells, NKT cells and dendritic cells show significant increases in co-expression of IFN-γ and IL-17 (Fig 4B, 4D–4F and 4H). However, the IFN-γ+ IL-17+ cells in the spleen are mostly neutrophils which do not exhibit increases in co-expression of these cytokines in dependent mice compared to non-dependent controls (Fig 4G and 4J). The observed cell-specific changes suggest that IFN-γ and IL-17 co-expression is relevant

in a variety of splenic cell types in alcohol dependent mice, but altogether does not amount to significantly higher levels of IFN-γ⁺ IL-17⁺ cells overall within the total CD45⁺ population in the spleen (Fig 4A).

In the liver, there were not significant differences in the percentage of IFN-γ⁺ IL-17⁺ cells within any population, although T cells, macrophages and neutrophils look to have increases which do not reach statistical significance (Fig 4D, 4F and 4G). A significant decrease in the NK cell proportion of CD45⁺ IFN-γ⁺ IL-17⁺ cells is accompanied by trends towards increased percentages of T cells, macrophages, and neutrophils in response to chronic alcohol exposure (Fig 4K). The expression levels in these populations largely varied across samples, so it is difficult to say there is a meaningful increase despite higher average values in dependent mice. T cells, macrophages and neutrophils may complement each other in contributing to the overall increase of IFN-γ and IL-17 co-expressing cells in dependent mice: T cells are the most highly expressed cell type in the liver but a lower percentage are IFN-γ⁺ IL-17⁺ compared to macrophages and neutrophils; macrophages and neutrophils are less populous in the liver but have a higher percentage are IFN-γ⁺ IL-17⁺ compared to T cells (Figs 1C and 4D, 4F and 4G).

## Discussion

In the present study, we found that alcohol dependent mice express higher levels of IFN-γ and IL-17 compared to non-dependent controls in their blood, spleen and liver. The data suggest that neutrophils and B cells are the major contributors to systemic pro-inflammatory type 1 and type 17 cytokine responses in alcohol dependent mice, although the importance of individual cell types varies in an organ-specific and cytokine-specific manner.

In the blood and spleen, neutrophils and B cells are primarily responsible for increased IFN-γ and IL-17 expression in alcohol dependent mice. Blood NK cells and splenic macrophages are additional contributors to increased IL-17 and IFN-γ expression, respectively. In the liver, we did not find any cell types to be majorly responsible for increased cytokine expression, but we hypothesize that non-significant trends towards increases in many cell types have cumulative effects on overall IFN-γ and IL-17 levels in alcohol dependent mice. While trends towards increases in IFN-γ and IL-17 individually in the liver are not significant, there is a significant increase in the overall co-expression of these cytokines due to T cells, macrophages and neutrophils. We also observed increased co-expression of IFN-γ and IL-17 in blood neutrophils.

Our analysis reveals mechanisms through which cytokine expression by various cells promotes systemic type 1 and type 17 immune responses in alcohol dependent mice. Some of these inflammatory mechanisms have been addressed previously using *in vitro* or *in vivo* models of chronic alcohol exposure, but no other studies have investigated a similarly large range of cytokines, cell types and organs simultaneously [9–12,15–17].

Notably, in this study we used a robust animal model of alcohol dependence that achieves reliable high blood alcohol levels, importantly mimicking the high blood alcohol levels in humans with AUD, enabling the study of dependence-induced neuroadaptations rather than effects of acute intoxication [19,20,22,23]. Most analyze one specific inflammatory process and focus on its role in the development of alcoholic liver disease (ALD), a leading cause of alcohol-related deaths [28]. In our analysis, we observed dynamic inflammatory processes which occur independent of ALD and identified the key cellular and molecular mediators of this systemic inflammation.

We observed an increase in IFN-γ expression but no change in IL-4 expression and concluded that chronic intermittent alcohol exposure in mice skews immunity towards systemic type 1 responses. The literature provides evidence both in support and opposition of these

results. Excessive TNF-α production—evidence of type 1 responses—and increased IgE levels —evidence of type 2 responses—are hallmarks of alcoholism [4,7,29]. Studies of IFN-γ and IL-4 expression provide less consistent conclusions. Some data show increased IFN-γ/IL-4 ratios whereas others find the opposite [4,10,11,30–32]. Based on this, it is still unclear how IFN-γ and IL-4 production is altered in response to chronic alcohol use.

Although many of these studies report T cell cytokine secretion, we observed that T cells are not the main drivers of differential cytokine responses in alcohol dependent mice (S1 Fig). In fact, for every cell type besides T cells we saw significant differences in IFN-γ expression in at least one organ. B cells, macrophages, and neutrophils most commonly expressed IFN-γ at higher levels in alcohol dependent mice.

No studies of chronic alcohol use address IFN-γ production by any of these cell types, although there is limited evidence that B cells and neutrophils produce IFN-γ in response to infection or stimulation [33–41]. Macrophages are known to secrete IFN-γ in vitro, and lung-resident alveolar macrophages have also been described to secrete IFN-γ in response to pulmonary infection in vivo [42–46]. Experiments with T cells may overlook meaningful alterations in type 1 and type 2 immunity due to cytokine production by other cells. Instead, studies should always analyze overall cytokine levels and consider neutrophils, B cells and macrophages as better measures of these changes.

We also observed systemic increases in IL-17 levels in alcohol dependent mice. Type 17 responses to chronic alcohol exposure are less studied than type 1 and type 2 responses and most research addresses their role in alcoholic liver disease (ALD) development. Patients with ALD have increased IL-17 levels in the blood and liver which correlate with disease severity. In these patients, IL-17+ liver-infiltrating cells are mostly T cells and neutrophils [9]. IL-17 blockade can reverse alcohol dependence and liver damage in mice [47]. Indeed, we found increases in neutrophil abundance and expression of IL-17 in the liver of alcohol dependent mice. Previous research found IL-17 is involved in neutrophil recruitment to the liver in ALD [9]. Our data support an additional ALD-independent role for IL-17+ neutrophil infiltration in the liver during the early stages of alcohol-induced inflammation.

Our data show IL-17 expression also increases in the blood and spleen in response to chronic intermittent alcohol exposure in mice. Systemic IL-17-mediated inflammation due to chronic alcohol exposure has not been investigated beyond the observation that IL-17 expression in the serum is increased in ALD patients [9]. We found that IL-17 expression in neutrophils, NK cells, and B cells increased dramatically in the blood of alcohol dependent mice compared to non-dependent controls. Furthermore, we are the first to report increased splenic B cell IL-17 expression that drives increases in spleen IL-17 levels.

It is well established that chronic alcohol exposure inhibits NK cell cytotoxic functions and modulates B cell antibody production [48–53]. Although neither cell type has been implicated in alcohol-induced IL-17 production, there is evidence that both can contribute substantially to type 17 responses in infection and autoimmune disorders [54–56].

The mechanisms behind altered IL-17 expression in alcoholics are not understood. Our data demonstrate that previously described increases in IL-17 in the liver of ALD patients are not exclusively a consequence of ALD: IL-17 expression increases systemically in the blood, spleen and liver after chronic intermittent alcohol exposure. These changes occur prior to any clinical manifestations of chronic alcohol use in mice. More studies are necessary to understand the molecular and cellular mechanisms which promote systemic type 17 responses to chronic alcohol exposure and the long-term effects on host health.

We also found increases in co-expression of IFN-γ and IL-17 contributed to the skewing of type 1 and type 17 responses. IFN-γ+ IL-17+ cells are not commonly studied, but there is evidence of T cell subsets which co-express these cytokines in chronic infections or autoimmune

disorders [57–61]. The mechanisms which promote IFN-γ[+] IL-17[+] T cells and their role in host defense is unclear. In dependent mice we did observe increases in IFN-γ[+] IL-17[+] T cells in the blood and spleen, as well as novel evidence of B cells, NKT cells, macrophages, neutrophils and dendritic cells with increased IFN-γ IL-17 co-expression in at least one organ. IFN-γ[+] IL-17[+] cells should be considered in future studies as a relevant component of systemic inflammation due to chronic alcohol use.

In this analysis, we give a snapshot of diverse immune processes which are altered in alcohol dependent mice. The study's strength lies in the wide range of immune processes investigated as we analyzed a variety of organs, cell types and cytokine secretion profiles to understand shifts in immune system characteristics in alcohol dependent mice. We did not perturb the immune system with other stimuli or pathogens, gaining insight into how immunity is altered at resting state.

In our model of alcohol dependence, mice were exposed to ethanol vapor intermittently for 4 weeks [18]. Of note this ethanol model is most commonly used for studies of addiction as it results in higher blood alcohol levels but does not induce the same levels of liver damage as alcohol drinking. Furthermore, rodent models that more closely mimic harmful inflammatory processes like ALD in humans may use an alcohol exposure time of 2–3 months [62]. The purpose of this study was not to understand ALD, rather to describe the systemic inflammatory processes that occur as mice develop dependence on alcohol. We cannot rule out that the inflammatory processes observed here lead directly to liver damage. Regardless, altered type 1 and type 17 responses are worth further investigation for their role in altered systemic immunity and initiation of potentially harmful inflammation.

Our main goal was to investigate as many immune processes as possible which limited precision in differentiating between cell types. In distinguishing macrophage, neutrophil and dendritic cell populations, we may have misclassified some of the rarer populations. Our phenotypic definition of neutrophils excluded CD11c[+] neutrophils and that of macrophages excluded F4/80- macrophages. F4/80- CD11c- macrophages specifically would be identified as neutrophils in our analysis (S1 Fig). We did not have the neutrophil-specific marker Ly6G included in the experiment which would have provided a clearer way to distinguish these populations [63].

Furthermore, defining type 1 and type 2 responses by only IFN-γ and IL-4 simplifies these dynamic processes. We did not observe an increase in IL-12, a stimulator of type 1 responses, in dependent mice despite higher IFN-γ levels [27]. It would have been beneficial to compare our results to previous studies if we had measurements of type 1 and type 2 markers TNF-α and IgE, respectively, which are commonly seen to be elevated in models of alcohol dependence. Nevertheless, we observed clear increases in systemic IFN-γ and IL-17 expression and believe future studies of chronic alcohol use would benefit from including these cytokines in their analyses.

The data provide insight into dynamic systemic responses that are underappreciated in alcohol research. Through our approach, we observed diverse immune processes which are altered in alcohol dependent mice. Future studies will assess whether different models of chronic alcohol exposure in mice such as liquid ethanol diet or a longer time period of alcohol exposure would provide a more human-like pattern of alcohol consumption and be more physiologically relevant to how alcohol affects the human body. In addition, a time course experiment analyzing the expression of inflammatory cytokines across the disease progression, including at an acute single alcohol intoxication, would be particularly insightful.

While most studies address the effects of chronic alcohol consumption on ALD, our data highlight systemic inflammatory processes which are activated prior to liver disease. These systemic changes could be important initiators of the severe long-term effects where the field

currently focuses. We identify neutrophils and B cells as substantial contributors to early inflammatory processes of chronic alcohol exposure. Future studies are needed for a better understanding of mechanisms through which neutrophils and B cells are activated and how their altered functionalities contribute to the adverse effects of AUD.

In conclusion, we found that chronic intermittent alcohol exposure in mice induces systemic IFN-γ and IL-17-mediated inflammatory responses. A variety of cell types are responsible for these responses in a cytokine- and organ-specific manner, but neutrophil and B cell cytokine secretion patterns are the most commonly dysregulated across all organs studied. We observed these changes *in vivo* without additional stimulation suggesting that alcohol dependence alters immune system functions at steady state. The data we present provide valuable insight into systemic inflammatory responses in alcohol dependent mice and serve as a starting point for future studies to probe these alcohol-induced inflammatory mechanisms.

## Supporting information

**S1 Fig. Gating strategy to identify individual cell types isolated from blood, spleen and liver of dependent and non-dependent control mice [63].**
(TIF)

**S2 Fig. Alcohol dependent mice exhibit minor tissue specific changes in their T cell subset ratios.** (A) Th1 (CD4$^+$ IFN-γ$^+$), Th2 (CD4$^+$ IL-4$^+$), Th17 (CD4$^+$ IL-17$^+$), and Treg (CD4$^+$ FoxP3$^+$) expression as a percentage of CD4$^+$ T helper cells isolated from spleen of alcohol dependent (red, solid fill) and non-dependent (black outline, white fill) mice. (B-C) CD4$^+$ and CD8$^+$ T cell expression as a percentage of CD3$^+$ cells isolated from (B) liver and (C) blood of dependent and non-dependent mice. (D) Th1, Th2, Th17 and Treg expression as a percentage of CD4$^+$ T helper cells isolated from blood of alcohol dependent and non-dependent mice. $^*$, p<0.05; $^{**}$, p<0.01 analyzed by Mann-Whitney U test; n = 5–8.
(TIF)

**S3 Fig. IL-10 receptor expression is increased in alcohol dependent mice compared to controls.** (A) Expression of IL-10 receptor (IL-10R) blood, spleen and liver CD45$^+$ cells isolated from alcohol dependent (red, solid fill) and non-dependent (black outline, white fill) mice. (B-D) Immune cells as percentages of all IL-10R$^+$ CD45$^+$ cells in (B) blood, (C) spleen and (D) liver isolated from dependent and non-dependent mice. (E-K) Expression of IL-10R by (E) B cells, (F) NK cells, (G) T cells, (H) NKT cells, (I) macrophages, (J) neutrophils and (K) dendritic cells in dependent and non-dependent mice. $^*$, p<0.05; $^{**}$, p<0.01 analyzed by Mann-Whitney U test; n = 5–8.
(TIF)

**S4 Fig. PD-1 expression is minimally altered in lymphocytes of alcohol dependent mice compared to controls.** (A-D) Expression of PD-1 by (E) B cells, (F) NK cells, (G) T cells, and (H) NKT cells in alcohol dependent (red, solid fill) and non-dependent (black outline, white fill) mice. $^*$, p<0.05; $^{**}$, p<0.01 analyzed by Mann-Whitney U test; n = 5–8.
(TIF)

**S1 Table. Antibodies used for the multiparametric flow cytometry experiment.**
(TIF)

## Author Contributions

**Conceptualization:** Silke Paust.

**Formal analysis:** Kayla Frank, Silke Paust.

**Funding acquisition:** Marisa Roberto, Silke Paust.

**Investigation:** Kayla Frank, Shawn Abeynaike, Rana Nikzad, Reesha R. Patel, Amanda J. Roberts.

**Methodology:** Shawn Abeynaike, Rana Nikzad, Reesha R. Patel, Amanda J. Roberts, Marisa Roberto, Silke Paust.

**Project administration:** Amanda J. Roberts, Marisa Roberto, Silke Paust.

**Resources:** Amanda J. Roberts, Marisa Roberto, Silke Paust.

**Supervision:** Silke Paust.

**Validation:** Reesha R. Patel, Marisa Roberto, Silke Paust.

**Visualization:** Kayla Frank, Silke Paust.

**Writing – original draft:** Kayla Frank.

**Writing – review & editing:** Reesha R. Patel, Marisa Roberto, Silke Paust.

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
