## [Decision Letter · Decision Letter 0]

9 Oct 2020

PONE-D-20-27238

Alcohol dependence promotes systemic IFN-γ and IL-17 responses in mice

PLOS ONE

Dear Dr. Paust,

Thank you for submitting your manuscript to PLOS ONE. After careful consideration, we feel that it has merit but does not fully meet PLOS ONE’s publication criteria as it currently stands. Therefore, we invite you to submit a revised version of the manuscript that addresses the points raised during the review process.

We look forward to receiving your revised manuscript.

Kind regards,

Junpeng Wang, Ph.D

Academic Editor

PLOS ONE

Journal Requirements:

2. To comply with PLOS ONE submissions requirements, please provide methods of sacrifice in the Methods section of your manuscript.

3. Please expand the acronym “TSRI” (as indicated in your financial disclosure) so that it states the name of your funders in full.

*This information should be included in your cover letter; we will change the financial disclosure on the online submission form on your behalf.*

Reviewers' comments:

Reviewer's Responses to Questions

**Comments to the Author**

1. Is the manuscript technically sound, and do the data support the conclusions?

Reviewer #1: Partly

Reviewer #2: Yes

2. Has the statistical analysis been performed appropriately and rigorously? 

Reviewer #1: Yes

Reviewer #2: Yes

3. Have the authors made all data underlying the findings in their manuscript fully available?

Reviewer #1: Yes

Reviewer #2: Yes

4. Is the manuscript presented in an intelligible fashion and written in standard English?

Reviewer #1: Yes

Reviewer #2: Yes

5. Review Comments to the Author

Reviewer #1: PONE-D-2027238

Overall: This is a nice brief report assessing immune cell populations and their associated cytokines during alcohol intoxication. It adds to the literature by assessing multiple tissues simultaneously and assessing for cell type origin of important mediators such as IFNg and IL-17. The manuscript is technically sound piece of scientific research with data that mostly supports the conclusions. However, the authors state these are changes due to their dependence model, though it cannot be determined if these are dependence-related changes or intoxication-related changes, since a single vapor exposure group is not included and dependent mice were sacrificed after a16h ethanol vapor session. Adding this important treatment group would help the field to understand acute vs chronic peripheral immune modulations in response to ethanol exposure. Otherwise, this is a nice and straight forward study that investigates cellular changes in IFNg and IL-17 expression in various tissues with ethanol treatment.

Results:

1. Why were IL10tm1FLv mice used? Was this an attempt to obtain accurate IL-10 labeling? This should be noted in the text

2. It is important to emphasize that these findings are during intoxication as mice were sacrificed directly from the vapor chambers and cytokine changes vary during intoxication and withdrawal.

3. The gating scheme appears mostly correct. However, were T cells and NK cells gated from CD11b negative and CD11c negative populations? This is not clear from the table. This could be enhanced by a graphical example with sample flow data.

4. Have circulating endotoxin levels been measured with this vapor model?

5. Can the authors measure IFNg and IL-17 in serum by ELISA or Luminex?

6. Figure 1: subtle changes in T cells in blood and spleen, though reduced CD4 T cells seen in the spleen which should be in a main figure, along with the Th1/Th2 imbalance in the liver.

a. Neutropenia is consistent with previous reports

b. These changes could represent a migration of T cells from circulation (spleen represents circulating populations) into the liver.

7. The % of total IFNg+ CD45cells was increased in blood/spleen which is important. However a specific subtype of IFNg+ cells was not identified. Was this because IFNg was increased slightly (non-significant) across multiple cell types e.g. macrophages, neutrophils, and dendritic cells (Fig 2B)? In other words, if statistics are done on those 3 cell populations (e.g. ANOVA) is there an etoh effect?

8. Figure 2 can be a bit confusing to the reader as presented. I recommend placing panels D-J first (making them A-G) and having the data in A-C last as H-J. This way it is clear to the reader that IFNg is increased in B Cells, NK cells and blood neutrophils etc. Then the reader will see the relative distributions across cell types with the other data. As presented it is easy to be confused.

9. Figure 3 – same organizational comment as for figure 2

Discussion:

1. It is unclear from this study if this is a feature of alcohol dependence or if it is a feature of acute alcohol intoxication. In order to make the claim that these are dependence associated changes, a group of mice that just receive the single vapor exposure (16h) prior to sacrifice is needed.

Pertinent references to include and mention:

1. Pasala et al 2015, PMID: 26695744

2. Gonzalez-Reimers et al 2012, PMID: 22510812

Reviewer #2: The manuscript titled “Alcohol dependence promotes systemic IFN-γ and IL-17 responses in mice” submitted by Fran K et al, evaluated the systemic immune modulations associated with the alcohol abuse and concluded that chronic alcohol consumption increases the pro-inflammatory responses. Although the pro-inflammatory effects of alcohol abuse are well established, this study was performed under non-pathological conditions. The overall manuscript is well written and I have the minor concerns.

1. Discussion and the result sections are redundant and the overall discussion is very lengthy. Moreover, the authors did not comment on the mechanistic insight into their findings. How this chronic inflammation may contribute to the pathologies associated with the organ systems they analyzed.

2. It is better to include the fluorescent plots to demonstrate the gating strategy instead of Table 2.

6. PLOS authors have the option to publish the peer review history of their article (what does this mean?). If published, this will include your full peer review and any attached files.

Reviewer #1: No

Reviewer #2: No

---

## [Author Response · Author response to Decision Letter 0]

23 Nov 2020

Dear Dr. Wang and the reviewers,

We would like to thank the editor and the reviewers for their thoughtful review of our manuscript. We have addressed each point below and made the corresponding changes in the figures and text. We feel that the revisions have greatly strengthened the manuscript and are looking forward to your response. We have addressed each point raised by the editor and reviewer below “Response:”.

Journal Requirements:

Response: We have reformatted the manuscript to meet the PLOS ONE’s style requirements.

2. To comply with PLOS ONE submissions requirements, please provide methods of sacrifice in the Methods section of your manuscript.

Response: We have added this information to the methods section.

3. Please expand the acronym “TSRI” (as indicated in your financial disclosure) so that it states the name of your funders in full.

*This information should be included in your cover letter; we will change the financial disclosure on the online submission form on your behalf.*

Response: We have added this information as requested.

Review Comments to the Author

Reviewer #1: PONE-D-2027238

Overall: This is a nice brief report assessing immune cell populations and their associated cytokines during alcohol intoxication. It adds to the literature by assessing multiple tissues simultaneously and assessing for cell type origin of important mediators such as IFNg and IL-17. The manuscript is technically sound piece of scientific research with data that mostly supports the conclusions. However, the authors state these are changes due to their dependence model, though it cannot be determined if these are dependence-related changes or intoxication-related changes, since a single vapor exposure group is not included and dependent mice were sacrificed after a16h ethanol vapor session. Adding this important treatment group would help the field to understand acute vs chronic peripheral immune modulations in response to ethanol exposure. Otherwise, this is a nice and straight forward study that investigates cellular changes in IFNg and IL-17 expression in various tissues with ethanol treatment.

Response: We thank the reviewer for commenting on the differences between alcohol dependence versus acute intoxication models. We have re-phrased these sections of the manuscript and specify that the dependent mice were euthanized while still intoxicated (Materials and Methods - end of page 6) and added, to the discussion, that future studies including acute alcohol intoxication time points would be valuable (page 22).

It is important to emphasize that the Roberto laboratory has extensively used the chronic ethanol exposure model, also utilized for this study, to induce ethanol dependence in mice (Herman et al., 2016, Schweitzer et al., 2016, Agoglia et al., 2020; Patel et al., 2019, Roberts AJ. 2019 Brain Behav. Immun on IL6; Bajo et al., 2019, Varodayan, Sidhu et al., Neuroph 2018, PMID: 27798128 

PMID: 26946429, PMID: 30791967, PMID: 32041742, PMID: 31437534, PMID: 29471053) and rats (Roberto et al., 2010, Herman and Roberto, 2016; Khom et al., JN 2020, (PMID: 25170988, PMID: 32769108, PMID: 20060104). This chronic 4-6 weeks long ethanol exposure always produces reliable high blood alcohol levels. It also induces an ethanol dependent state enabling us to identify profound neuroadaptation in synaptic transmission in several brain regions as well as changes in several neuroimmune signaling (microglia, IL1beta etc) (Warden et al., 2020; Patel et al., 2019, PMID: 32680583, PMID: 30791967). To build on this large body of studies conducted in the brain, this study's main goal was to determine the effect of chronic ethanol exposure (for four weeks) on several other peripheral organs. Given that chronic alcohol alters inflammatory processes that regulate immune function, the present results are very important for understanding the complex immune defense mechanisms that promote inflammation and may contribute to the development of alcoholism. As described in the introduction and discussion, there is a lack of studies addressing the interplay between organ-specific inflammatory response mechanisms of individual cell types to chronic alcohol exposure at a systemic level. 

Results:

1. Why were IL10tm1FLv mice used? Was this an attempt to obtain accurate IL-10 labeling? This should be noted in the text.

Response: Yes, and we added this information and a reference in the Materials and Methods, Animals section.

2. It is important to emphasize that these findings are during intoxication as mice were sacrificed directly from the vapor chambers and cytokine changes vary during intoxication and withdrawal.

Response: We thank the reviewer for their helpful comment on this important aspect of our study. We have added this information as requested in the revised materials and methods and discussion sections.

3. The gating scheme appears mostly correct. However, were T cells and NK cells gated from CD11b negative and CD11c negative populations? This is not clear from the table. This could be enhanced by a graphical example with sample flow data.

Response: We have added this information as requested as a new supplemental figure 1 which shows the gating strategy.

4. Have circulating endotoxin levels been measured with this vapor model?

Response: Unfortunately, we did not separate peripheral blood mononuclear cells from sera but used an anticoagulant and a gradient centrifugation to isolate PBMC and as such do not have sera for analyses. As such, we did not measure circulating endotoxin levels. 

5. Can the authors measure IFNg and IL-17 in serum by ELISA or Luminex?

Response: Unfortunately, we did not separate peripheral blood mononuclear cells from sera but used an anticoagulant and a gradient centrifugation to isolate PBMC and as such do not have sera for analyses. We will keep this helpful suggestion in mind for future experiments.

6. Figure 1: subtle changes in T cells in blood and spleen, though reduced CD4 T cells seen in the spleen which should be in a main figure, along with the Th1/Th2 imbalance in the liver.

a. Neutropenia is consistent with previous reports

b. These changes could represent a migration of T cells from circulation (spleen represents circulating populations) into the liver.

Response: We thank the reviewer for their helpful suggestions. We revised the figures as proposed by the reviewer.

7. The % of total IFNg+ CD45cells was increased in blood/spleen which is important. However a specific subtype of IFNg+ cells was not identified. Was this because IFNg was increased slightly (non-significant) across multiple cell types e.g. macrophages, neutrophils, and dendritic cells (Fig 2B)? In other words, if statistics are done on those 3 cell populations (e.g. ANOVA) is there an etoh effect?

Response: We did find statistically significant increases in IFN-y in specific cell types in the blood and spleen. We present and discuss these findings in the revised manuscript’s results section. In the blood of alcohol dependent mice, B cells and neutrophils have increased I IFN-y expression which we hypothesize (line 184-185) to be the source of overall increased blood IFN-y levels. In the spleen, we find significant increases in B cell and macrophage IFN-y levels in addition to increased abundance of neutrophils which we hypothesize (line 202-204) to be responsible for increased splenic IFN-y levels in alcohol dependent mice. 

The Analysis of macrophages, neutrophils, and dendritic cells by ANOVA did not reveal a statistically significant alcohol effect on IFN-y expression in these cell types in the blood or the spleen (blood, p=.1661; spleen, p=0.07570). This is likely because B cell upregulation of IFNy in alcohol dependent mice is largely contributing to overall increases in IFNy levels in these organs.

In the liver, the overall increase in IFN-y is not significant. Macrophages, neutrophils and DCs all slightly increase their IFN-y production, however, this is not statistically significant by ANOVA (p=0.0710)

8. Figure 2 can be a bit confusing to the reader as presented. I recommend placing panels D-J first (making them A-G) and having the data in A-C last as H-J. This way it is clear to the reader that IFNg is increased in B Cells, NK cells and blood neutrophils etc. Then the reader will see the relative distributions across cell types with the other data. As presented, it is easy to be confused.

Response: We thank the reviewer for these helpful suggestions, and we revised the figures as proposed by the reviewer.

9. Figure 3 – same organizational comment as for figure 2

Response: We revised the figures as proposed by the reviewer.

Discussion:

1. It is unclear from this study if this is a feature of alcohol dependence or if it is a feature of acute alcohol intoxication. In order to make the claim that these are dependence associated changes, a group of mice that just receive the single vapor exposure (16h) prior to sacrifice is needed.

Response: Please see our previous response to “Overall”. 

Also, we added the following relevant references to the revised discussion: 1. Pasala et al 2015, PMID: 26695744 2. Gonzalez-Reimers et al 2012, PMID: 22510812. The discussion now includes the following additional text: “Notably, in this study we used a robust animal model of alcohol dependence that achieves reliable high blood alcohol levels, importantly mimicking the high blood alcohol levels in humans with AUD, enabling the study of dependence-induced neuroadaptations rather than effects of acute intoxication [19, 20, 22, 23]” and “In addition, a time course experiment analyzing the expression of inflammatory cytokines across the disease progression, including at an acute single alcohol intoxication, would be particularly insightful.” 

Reviewer #2: The manuscript titled “Alcohol dependence promotes systemic IFN-γ and IL-17 responses in mice” submitted by Fran K et al, evaluated the systemic immune modulations associated with the alcohol abuse and concluded that chronic alcohol consumption increases the pro-inflammatory responses. Although the pro-inflammatory effects of alcohol abuse are well established, this study was performed under non-pathological conditions. The overall manuscript is well written, and I have the minor concerns.

1. Discussion and the result sections are redundant, and the overall discussion is very lengthy. Moreover, the authors did not comment on the mechanistic insight into their findings. How this chronic inflammation may contribute to the pathologies associated with the organ systems they analyzed.

Response: We thank the reviewer for their helpful comments and have shortened the discussion section to be less redundant.

2. It is better to include the fluorescent plots to demonstrate the gating strategy instead of Table 2.

Response: We have added this information as requested as a new supplemental figure 1.

---

## [Decision Letter · Decision Letter 1]

8 Dec 2020

Alcohol dependence promotes systemic IFN-γ and IL-17 responses in mice

PONE-D-20-27238R1

Dear Dr. Paust,

We’re pleased to inform you that your manuscript has been judged scientifically suitable for publication and will be formally accepted for publication once it meets all outstanding technical requirements.

Kind regards,

Junpeng Wang, Ph.D

Academic Editor

PLOS ONE

Additional Editor Comments (optional):

Reviewers' comments:

Reviewer's Responses to Questions

**Comments to the Author**

1. If the authors have adequately addressed your comments raised in a previous round of review and you feel that this manuscript is now acceptable for publication, you may indicate that here to bypass the “Comments to the Author” section, enter your conflict of interest statement in the “Confidential to Editor” section, and submit your "Accept" recommendation.

Reviewer #1: All comments have been addressed

2. Is the manuscript technically sound, and do the data support the conclusions?

Reviewer #1: Yes

3. Has the statistical analysis been performed appropriately and rigorously? 

Reviewer #1: Yes

4. Have the authors made all data underlying the findings in their manuscript fully available?

Reviewer #1: Yes

5. Is the manuscript presented in an intelligible fashion and written in standard English?

Reviewer #1: Yes

6. Review Comments to the Author

Reviewer #1: (No Response)

7. PLOS authors have the option to publish the peer review history of their article (what does this mean?). If published, this will include your full peer review and any attached files.

Reviewer #1: No

---

## [Editor Report · Acceptance letter]

11 Dec 2020

PONE-D-20-27238R1 

Alcohol dependence promotes systemic IFN-γ and IL-17 responses in mice 

Dear Dr. Paust:

I'm pleased to inform you that your manuscript has been deemed suitable for publication in PLOS ONE. Congratulations! Your manuscript is now with our production department. 

Kind regards, 

on behalf of

Dr. Junpeng Wang 

Academic Editor

PLOS ONE